# A Health Belief Model-Based Motivational Interviewing for Medication Adherence and Treatment Success in Pulmonary Tuberculosis Patients

**DOI:** 10.3390/ijerph182413238

**Published:** 2021-12-15

**Authors:** Ni Made Parwati, I Made Bakta, Pande Putu Januraga, I Made Ady Wirawan

**Affiliations:** 1Doctoral Study Program, Medical Faculty, Udayana University, Denpasar 80361, Indonesia; 2Department of Internal Medicine, Medical Faculty, Udayana University, Denpasar 80234, Indonesia; madebakta@unud.ac.id; 3School of Public Health, Udayana University, Denpasar 80234, Indonesia; januraga@unud.ac.id (P.P.J.); ady.wirawan@unud.ac.id (I.M.A.W.)

**Keywords:** motivational interviewing, Health Belief Models, tuberculosis

## Abstract

Medication adherence behavior plays a central role in the success of tuberculosis (TB) treatment. Conventional motivation is not optimal in strengthening long-term medication adherence. A motivational interviewing (MI) communication motivation model based on the Health Belief Model (HBM) was designed with the main objective of improving medication adherence and treatment success. This study used an experimental design with a randomized posttest-only control group design. The intervention and control groups consisted of 107 TB patients each, who were selected by random cluster sampling. The study was conducted from November 2020 to June 2021 at 38 public health centers in Bali Province. The HBM-based MI model intervention was given in seven counseling sessions, pill count percentages were used to measure medication adherence, and treatment success was based on sputum examination results. Logistic regression was used to assess the effect of the intervention on medication adherence and treatment success. Logistic regression analysis showed that MI-based HBM and knowledge were the most influential variables for increasing medication adherence and treatment success. Medication adherence was 4.5 times greater (ARR = 4.51, *p* = 0.018) and treatment success was 3.8 times greater (ARR = 3.81, *p* < 0.038) in the intervention group compared to the control group, while the secondary outcome of knowledge of other factors together influenced medication adherence and treatment success. The conclusion is that the HBM-based MI communication motivation model creates a patient-centered relationship by overcoming the triggers of treatment barriers originating from the HBM construct, effectively increasing medication adherence and treatment success for TB patients, and it needs further development by involving families in counseling for consistent self-efficacy of patients in long-term treatment.

## 1. Introduction

Tuberculosis is a global public health challenge. Medication adherence behavior plays a central role in the success of TB treatment [1]. In Indonesia, the Stop TB Strategy for the control of pulmonary TB with the directly observed treatment short course (DOTS) mechanism has been implemented since 1995. Community and health workers are trained with the aim of removing some of the barriers to medication adherence. In addition to interventions with conventional motivations that have not been optimally strengthened for long-term medication adherence, there were 9073 cases of multi-drug resistant (MDR) TB in 2018, confirmed to be the result of noncompliance, further resulting in suboptimal medical efforts, which endangered patient health and led to high morbidity and mortality rates [2].

Medication adherence and treatment success requires a multifaceted approach in order to help adapt, change, and maintain behavior due to the long treatment process of pulmonary TB. An approach is needed with the concept of individual-centered care to balance the rights and needs of patients, including the responsibility for their recovery. One model used to explain and understand health behavior in TB treatment adherence is the Health Belief Model (HBM) [1,3].

The motivational interviewing (MI) communication model based on the Health Belief Model (HBM) is used to strengthen intrinsic motivation and increase intrinsic motivation to change through understanding and resolving ambivalence between current behavior and future goals and values. Adherence behavior in treatment will occur if individuals perceive themselves as potentially vulnerable (susceptible), view the disease as severe, are convinced that the prevention regimen is effective (benefits), and see that there are some difficulties in or barriers to achieving recovery [4,5].

It is important to develop a new approach related to TB patients’ belief in healthy behavior (health belief) to improve medication adherence and treatment success by integrating HBM and MI-based health education models on an ongoing basis. This education starts from the view that individual behavior is influenced by psychological impulses, and MI is more focused on efforts to strengthen self-efficacy, which gives patients confidence and enthusiasm to undergo long-term treatment [6].

This study aims to evaluate medication adherence in terms of dose accuracy, time and duration of treatment, and the success of treatment based on negative results of acid-resistant bacteria sputum examination after counseling with the motivational interviewing model approach based on the Health Belief Model.

## 2. Materials and Methods

### 2.1. Theory Used in the Motivational Interviewing Model

Counseling with an HBM-based motivational interviewing approach was used to assist pulmonary TB patients in identifying triggers for specific barriers in the treatment process. The assumption underlying the HBM is that health behavior is motivated by beliefs, how individuals feel about the threat of disease. The HBM suggests that health-related behavior depends on the person’s desire to avoid disease and the belief that certain healthy actions will prevent disease. Patients with pulmonary TB will have treatment success if they have a level of motivation and relevant health knowledge [7]. According to the HBM concept, in the management of pulmonary TB treatment, self-efficacy can be increased through knowledge along with support from health/social workers and medication supervisors so as to promote obedience in treatment during the counseling process [4]. The HBM theory underlies the counselor’s actions in identifying barriers to medication adherence and treatment success based on the constructs of perceived susceptibility and disease severity (perceived severity), triggers of barriers related to treatment that are not useful (perceived benefits), and triggers of low self-efficacy regarding the ability to undergo complete treatment, including identifying triggers for interference in cues to action [8]. From the results of identifying obstacle triggers, counselors communicate and provide motivation using the MI approach to help create plans with the aim of changing the triggers toward behavioral changes that support drug adherence and treatment success.

### 2.2. Study Design and Participants

This study used an experimental design with a randomized post-test only control group design with 214 pulmonary tuberculosis patients at 38 public health centers in Bali Province from November 2020 to June 2021. TB counselors provided counseling using the motivational interviewing communication model based on the Health Belief Model, scheduled as one counseling session per week by telephone or online media, with a maximum of 7 counseling sessions in 2 months. Eligible patients in this study were (1) new patients with an early diagnosis of pulmonary TB by a molecular rapid test that had been recorded in the Tuberculosis patient register at the community health center and who had never received anti-tuberculosis treatment, and (2) adult patients aged between 18–65 years to facilitate monitoring during the study. Patients who did not meet the requirements were (1) patients with multi-drug resistant drug-resistant TB (MDR TB) as recorded in the TB register, (2) patients with other comorbidities based on a doctor’s examination that could interfere with the ability to assess adherence of anti-tuberculosis drugs without being influenced by other drugs, such as diabetes mellitus, liver cirrhosis, hepatitis, or cardiovascular disease, and (3) patients who were not willing to undergo tuberculosis treatment until the advanced phase (Figure 1).

### 2.3. Sampling and Data Collection

Determination of sample size to test the 2-proportion-difference hypothesis (1-sided test) was calculated considering that previous studies have shown 30% initial noncompliance in the intervention group; found: P1 = 25% and P2 = 10%, the researchers tested the hypothesis with a significance degree of 5% and a power of 80%. The sample size is calculated using the following formula [1,9]: $$\frac{n={\left(Z1-\alpha /2\surd 2\left(1-P\right)+Z1-\beta \surd P1\left(1-P\right)+P2\left(1-P2\right)\right)}^{2}}{{\left(P1-P2\right)}^{2}}$$

This calculation obtained a sample size of 97 for each arm, and sample size correction showed an estimated dropout of 10%, so the sample size of each arm was 107. The total sample used was 214 people.

The intervention and control groups were determined using a simple randomization technique. A total of 38 community health centers were selected and allocated to the intervention group (*n* = 19) and control group (*n* = 19) using random numbers generated from a computer program. The number of samples at each community health center was adjusted proportionally according to the target.

Data on new patients in the treatment and control groups were obtained from the TB patient register at the community health centers based on the inclusion criteria. Other data were collected by online questionnaires, including variables of income, knowledge, family support, and perceptions of the attitudes of health workers.

### 2.4. Primary Outcome

There were two primary outcomes:1.Improved medication adherence for 6 months in each counseling group. The results of the HBM-based MI intervention were compared with conventional counseling. Adherence to taking medication was measured by the percentage of pills taken. The number of unit doses taken by the patient in the second and sixth months of treatment was checked at the time of the visit to the community health center. This number was then compared with the total number of units the patient received to calculate the compliance ratio.

The number of pills taken at the end of the second and sixth months of treatment were calculated using the ratio between the number of drugs taken and the number of drugs to be taken (as a percentage). If there was overuse (calculation ratio > 100%), then the percentage of patient compliance was calculated from the ratio between the number of drugs consumed minus the number of excess drugs consumed and the number of drugs still to be consumed [10].

2.Treatment success was measured by bacteriological sputum examination, carried out after each phase of treatment. The examination was carried out following the standard reference for TB treatment, namely at the end of the intensive phase in the second month, and in the fifth and sixth months. Treatment was considered successful if the sputum was negative at the end of the treatment phase and at one of the previous examinations [11].

### 2.5. Secondary Results

The secondary outcome was determined based on the influence of other factors: age, education, income, knowledge, family support, and perceptions of the attitudes of health workers toward medication adherence and treatment success.

### 2.6. Intervention Procedure

The HBM-based MI intervention began in November 2020 and was carried out by telephone or video call. Before the intervention began, we developed a counselor training module and curriculum for health workers. In addition to providing training for counselors in the intervention group, a control group workshop on conventional motivation was also conducted and the number of each counselor between groups was 19 people from the TB program manager. HBM-based MI counseling was given to TB patients from the first contact with health workers. TB counselors are specially trained by an accredited health training institution in the Province of Bali. The training is carried out for 10 days, and the trainers have experience in training as TB counselor trainers. Counseling was given for two months, with a maximum of 7 sessions. Simultaneously, medication adherence was measured in the sixth month of the treatment process.

The initial step taken by the counselor during the first contact with a pulmonary TB patient was to identify the patient’s position based on the stages of behavioral change (pre-contemplation, contemplation, preparation, action, and maintenance) as the basis to conduct HBM-based MI counseling [12]. After finding out the patient’s needs at the behavior change stage, the counselor explored the patient’s attitudes, feelings, and thoughts—including future treatment plans, identifying triggers to barriers to adherence, and treatment success based on the HBM construct—and provided motivation according to counseling guidelines to minimize the triggers.

Counseling was given for 2 months, followed by sputum smear examination in the second, fifth, and sixth months of treatment to evaluate the success of treatment based on negative results [11].

### 2.7. Statistical Analysis

Descriptive analysis was conducted to describe each variable measured in the study, showing the mean, standard deviation, and relative frequency in both groups. The Kolmogorov–Smirnov test was used to assess the distribution of data or variables with a significance of *p* > 0.05, and Levene’s test was used to test the variance of data between groups with a significance of *p* > 0.05. For bivariate analysis, the chi-square test with a significance level of *p* < 0.05, independent *t*-test, or Mann–Whitney test was used to analyze differences in proportions and average drug adherence of the 2 groups. Logistic regression analysis with a significance of *p* < 0.05 was used to analyze the effect of the HBM-based MI communication motivation model intervention on medication adherence and treatment success, as well as to control for other independent variables [9].

### 2.8. Ethics Statement

This research was registered with ISRCTN on 12 December 2020 (registration number ISRCTN 18141422), and was approved by the Research Ethics Commission of the Faculty of Medicine, Udayana University/Sanglah Hospital Denpasar, Bali, Indonesia, on 8 October 2020 (approval number 2049/UN 14.2.2.VII. 14/LT/2020), and a research permit was obtained from the Bali Provincial Government (number 070/2175/IZIN-C/DISPMPT).

## 3. Results

A total of 629 participants were eligible to participate in this study. Among them, 375 did not meet the inclusion requirements and 40 refused to give consent; 214 respondents were willing to participate and met the inclusion criteria, and they were selected by proportional stratified random sampling. All participants gave informed consent and completed the entire study procedure. Adherence to medication was monitored with pill count percentages, and treatment success was based on the results of sputum examination in the second, fifth, and sixth months of treatment. The two groups were comparable overall: there were no differences in the characteristics of the respondents regarding gender, age, marital status, education, or occupation, each with *p* > 0.05. From the results of the independent *t*-test, intervention and control groups were equal in terms of age (*p* > 0.05) and were not comparable regarding income *(p* < 0.05) (Table 1).

### 3.1. Primary Outcome

The primary outcome of this study, after seven sessions of counseling over two months, was the effect of the HBM-based MI communication motivation model on medication adherence and treatment success in the intervention and control groups. The control group showed statistical significance (*p* = 0.023) and treatment success showed a statistically significant difference in the proportion (*p* = 0.038). The results of the Mann–Whitney U test showed that there was a significant difference in average adherence between groups. The HBM-based MI model intervention resulted in a significant difference in mean adherence; there were higher adherence scores in the intervention group than in the control group (*p* = 0.37) (Table 2).

The results of multivariate analysis with logistic regression after controlling for variables of knowledge, family support, and perception of the attitudes of health workers showed good knowledge variables, and the HBM-based MI model intervention had the most dominant and significant effect on increasing drug adherence, with better results Adjusted Risk Ratio (ARR = 4.51; *p* = 0.018) and the effect of proper knowledge (ARR = 2.93; *p* = 0.042); the intervention and knowledge variables had the most significant effect on treatment success Adjusted Risk Ratio (ARR = 3.81; 95% Confidence Interval (CI): 1.193–10.223; *p* = 0.038), and proper knowledge Adjusted Risk Ratio (ARR = 3.49; *p* = 0.022) (Table 3).

### 3.2. Secondary Outcome

From the overall results of the analysis—with the chi-square test showing knowledge as another factor that influences medication adherence and treatment success—it was found that the difference in the proportion of knowledge was statistically significant, with *p* = 0.015, Relative Risk (RR) = 1.113 (95% Confidence Interval (CI), = 1.003–1.235) for medication adherence and *p* = 0.007, Relative Risk (RR) = 1.121 (95% Confidence Interval (CI), = 1.012–1.243) for treatment success (Table 4).

## 4. Discussion

The results of this study prove that intervention with the motivational interviewing communication model based on the Health Belief Model has an influence on medication adherence and treatment success in patients with pulmonary tuberculosis.

This effect can be achieved by providing motivation as needed so that patients are able to overcome the triggers of treatment barriers originating from the HBM construct, so that correct thoughts and perceptions about treatment will arise. With a belief-based approach to healthy behavior, the intervention is based on the triggers of barriers that arise at the behavioral change stage. Counseling according to the HBM-based MI model was carried out for a maximum of seven sessions over two months, and was provided online in response to the COVID-19 pandemic. Health workers provide guidance and strong motivation to change and increase readiness for long-term obedience to taking medication, which has an effect on increasing motivation by resolving ambivalence/doubt, which then encourages patients to be ready, willing, and able to influence their self-confidence for change [5,13]. The HBM-based MI communication motivation model intervention was successful and effective in motivating patients to continue treatment until it was completed, leading to a tendency for decreased microbes and a minimal risk of recurrence.

Motivational interviewing as an intervention for clients has produced positive results in medication adherence and treatment success when there had been a decrease in adherence at each stage of behavior change [14]. This communication motivation model is an effective client-centered approach based on the theory, and a new strategy for tuberculosis patients undergoing long periods of treatment with various complex problems to generate consistent self-efficacy and overcome barriers to drug adherence, supporting treatment success [13,15].

The main principle of this HBM-based MI counseling approach is that clients have different needs and problems at each stage of change, so they need different kinds of treatment [16]. The core factors that are considered when implementing this counseling are identifying each individual’s specific triggers, and exploring and paying attention to the triggers that influence medication adherence based on the Health Belief Model regarding susceptibility/severity/benefits/barriers, cues to action, and self-efficacy [8,17].

The online platform component remains a challenge, and online MI deserves to be implemented and well-received. This approach has considerable potential to reduce costs, minimize counselor burden, and benefit clients [18,19]. The HBM-based MI communication motivation model as an innovative approach provides theory-based interventions in an effort to change behavior by integrating HBM theory and developing a systematic counseling framework that can maximize medication adherence and encourage treatment success in patients with pulmonary tuberculosis [20,21].

In this study, the intervention of the HBM-based MI communication motivation model with knowledge simultaneously affected medication adherence and treatment success. A change in behavior toward the acceptance of new behavior, or adoption through a process based on knowledge, awareness, and positive attitude (predisposing factors), will be longer-lasting than behavior that is not based on knowledge. Knowledge is the basis for understanding obstacles to successful treatment, so that self-efficacy in the ability to carry out the treatment will emerge [22,23].

MI still provides better compliance results compared to other strategies. The empathic counseling style of MI and the externalization of internal conflicts increase natural motivation by the use of techniques such as asking open-ended questions, reflective listening, summarizing, and preparing for change [24,25].

Motivational interviewing encourages health workers to resolve patients’ ambivalence as the key to change, because ambivalence is often a barrier to action. Patients are able to interpret the obstacles that arise from the effects of treatment and other social problems and will be optimistic that they can undergo treatment, and are able to understand that the treatment process requires support from within and from the social environment, as well as the intention and readiness to change, which fosters long-term self-efficacy to adhere to medication [26].

Health psychology theory is useful for predicting adherence and has a direct effect on treatment success. The effectiveness of the intervention can be increased based on a broader theory by incorporating behavioral change techniques that allow synergistic effects [27].

Bacteriological conversion indicates patients without inactivity, and can predict treatment success, and early bacteriological conversion is the key to success. The risk of failure in treatment is higher for patients who do not adhere to anti-tuberculosis drugs [28]. One reason for loss to follow-up is that patients have difficulty following treatment recommendations and are not disciplined in taking medications due to loss of interest over the lengthy treatment [29].

The HBM-based MI communication motivation model is intended to reduce perceived barriers. Some patients are not able to minimize the appearance of perceived triggers to treatment barriers, so they discontinue treatment. Cues to act will have a greater influence on behavior in situations where perceived threats and benefits are high and perceived barriers are low [30].

With the intervention of the HBM-based MI model, advice from others about the disease (external cues) can provide cues to act, and cues can also arise internally, such as side effects from medication, discomfort, or fatigue. Cues to act are among the triggers of behavior change [31].

In the HBM model, cues or triggers are needed to induce engagement in health behaviors. The intensity of cues required to induce action varies between individuals with perceived susceptibility, seriousness, usefulness, and inhibition [32].

### Limitations

This study attempts to monitor compliance with taking medication by comparing the actual with the expected number of pills remaining. However, pill counts may not always show noncompliance (e.g., patients may lose, hide, give away, or dispose of pills, or may dose at incorrect quantities and/or frequencies); this can be prevented further by the patient’s family actively supervising the taking of medication. 

## 5. Conclusions

The HBM-based MI communication motivation model requires ongoing counselor training for tuberculosis program managers in health facilities in order to be competent to explore the problems and needs of patients and create patient-centered relationships by helping them overcome triggers of treatment barriers originating from the HBM construct. To effectively improve medication adherence and treatment success for TB patients, further development is necessary by involving families in counseling for consistent self-efficacy.

## Figures and Tables

**Figure 1 ijerph-18-13238-f001:**
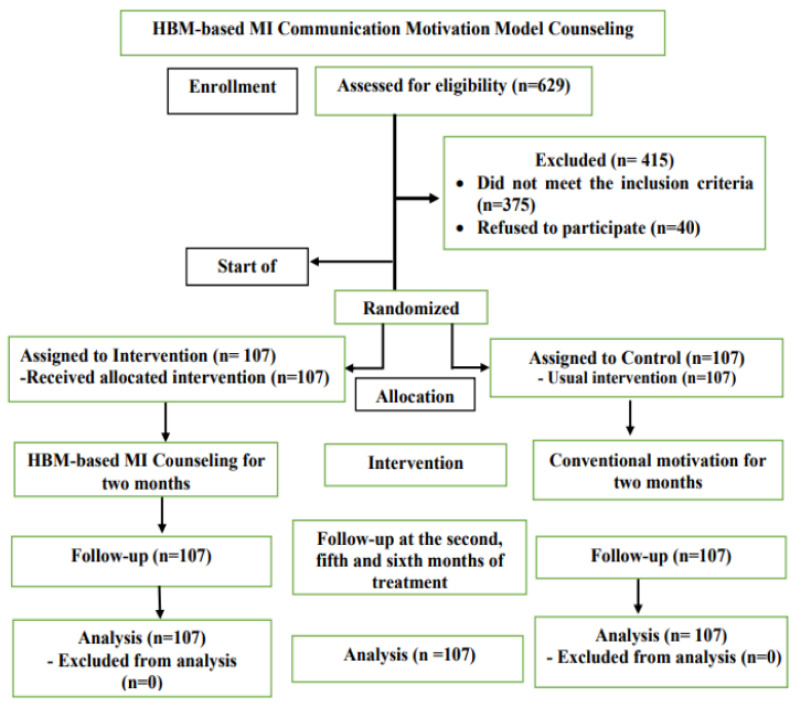
Consolidated Standards of Reporting Trials (CONSORT) diagram.

**Table 1 ijerph-18-13238-t001:** Between-group comparison of participants’ demographic characteristics.

	Characteristics	Intervention (*n* = 107)	Control (*n* = 107)	*p*-Value ^1^
1.	Gender			
	Male	64 (59.8)	60 (56.1)	0.580
	Female	43 (40.2)	47 (43.9)	
2.	Marital status			
	Married	89 (83.2)	82 (76.6)	0.087
	Unmarried	15 (14.0)	14 (13.1)	
	Other	3 (2.8)	11 (10.3)	
3.	Education level			
	Elementary or less Medium to high	56 (52.3) 51 (47.7)	67 (62.6) 40 (37.4)	0.128
4.	Have occupation			
	No	24 (22.4)	32 (29.9)	0.213
	Yes	83 (77.6)	75 (70.1)	
5.	Age (y)	42.3 ± 12	43 ± 10.9	0.412 ^2^
6.	Income *	2,310,280 ± 489,692	2,083,177 ± 100,000	0.000 ^2^

Values are presented as number (%) or mean ± standard deviation. ^1^ Chi Square test. ^2^ Independent-sample *t*-test. * In Rupiah (IDR).

**Table 2 ijerph-18-13238-t002:** Comparison of effects of HBM-based MI communication motivation counseling on medication adherence and treatment success between groups.

	HBM-Based MI Communication Motivation Model	Intervention	Control	*p*-Value
1	Adherence			
	Adherence	103 (96.3)	94 (87.9)	0.023 ^1^
	Non-adherence	4 (3.7)	13 (12.1)	
	Mean difference in adherence	115.65	99.35	0.037 ^2^
2	Treatment success			
	Success	103 (96.3)	95 (88.8)	0.038 ^1^
	No success	4 (3.7)	12 (11.2)	

Values are presented as number (%). ^1^ Results of Chi Square test. ^2^ Results of Mann–Whitney U test.

**Table 3 ijerph-18-13238-t003:** Dominant factors that influence medication adherence and treatment success.

	Variabel	Medication Adherence		Treatment Success	
		ARR *	*p*-Value *	ARR *	*p*-Value *
1.	MI-HBM intervention	4.51	0.018	3.81	0.038
2.	Good knowledge	2.93	0.042	3.49	0.022
3.	Good family support	0.63	0.507	0.708	0.618
4.	Good perception of health workers’ attitudes	0.46	0.233	0.538	0.337

* Result of Logistic regression analysis.

**Table 4 ijerph-18-13238-t004:** Influence of other factors on medication adherence and treatment success (*n* = 214).

	Other Variables	Medication Adherence						Treatment Success					
		Adherence		Non Adherence		*p*-Value ^1^		Success		Non Successful		*p*-Value ^1^	
1	Age (y)												
	18–45		107 (92.2)		9 (7.8)		0.913		107 (92.2)		9 (7.8)		0.913
	46–65		90 (91.8)		8 (8.2)				90 (91.8)		8 (8.2)		
2	Education level												
	Basic		112 (91.1)		11(8.9)		0.530		113 (91.9)		10 (8.1)		0.673
	Medium to high		85 (93.4)		6 (6.6)				85 (93.4)		6 (6.6)		
3	Income												
	Sufficient		54 (93.1)		4 (6.9)		0.730		54 (93.1)		4 (6.9)		0.844
	Insufficient		143 (91.7)		13 (8.3)				144 (92.3)		12 (7.7)		
4	Knowledge of TBC												
	Good		138 (95.2)		7 (4.80)		0.015		139 (95.9)		6 (4.1)		0.007
	Poor		59 (85.5)		10 (14.5)				59 (85.5)		10 (14.5)		
5	Family support												
	Good		157 (91.8)		14 (8.2)		0.240		158 (92.4)		13 (7.6)		0.189
	Poor		40 (93.0)		3 (7.0)				40 (93.0)		3 (7.0)		
6	Perceptions of attitudes of health workers												
	Good		146 (91.8)		13 (8.2)		0.188		147 (92.5)		12 (7.5)		0.147
	Poor		51(92.7)		4 (7.3)				51 (92.7)		4 (7.3)		

Values are presented as number (%). ^1^ Results of Chi Square test.

## Data Availability

Data will not be shared to protect the anonymity of the participants. Readers who wish to gain access to the data can write to the corresponding author, and data may be granted upon reasonable request.

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
