# Peer review of "A Health Belief Model-Based Motivational Interviewing for Medication Adherence and Treatment Success in Pulmonary Tuberculosis Patients"

_ijerph, 2021, doi:10.3390/ijerph182413238_

Round 1
Reviewer 1 Report
This Manuscript provides an interesting approach to patient adherence. I have specific comments I believe would add value to the Manusript.
- Did the TB counsellors undergo specific training to conduct these interviews?
- Please outline inclusion and exclusion criteria and rationales for them.
- Was there a sample size calculation?
- Please provide test used and how data are presented in table footnotes. Also define abbreviations.
- Was there difference among the interviewers? how many were there anyway?
- Was there difference regarding telephone or video call?
- Limitations should be added at the end of discussion section. i. e. pill counting method as some patients may have just tossed the pills down the drain etc.
Author Response
Dear reviewers 1,
For feedback points, please see the attachment.
Kind Regards,
Ni Made Parwati

Reviewer 2 Report
The text is interesting, because encourages counseling activities in order to reach therapeutic successes.
I suggest only one text editing:
- in materials and methods inclusion and exclusion criteria are not specified: it is appropriate to describe them.
Author Response
Dear reviewers 2,
For feedback points, please see the attachment.
Kind Regards
Ni Made Parwati

Reviewer 3 Report
In the current study, the authors present a health belief model-based motivational interviewing communication motivation model for medication adherence and treatment success in pulmonary tuberculosis patients. A randomized controlled trial was performed with 107 TB patients in intervention and control group respectively. The findings suggest that the implementation of the HBM-based MI communication model improved medication adherence and in turn the treatment success.
The article is well structured into section and subsections. It is within the scope of journal. However, the article needs to be improved for clarity to the readers.
There are comments to improve the article are as follows:
- Page 1, line 2-5: The title needs to short and precise.
Following are some suggestions: A motivational interviewing communication motivation model for medication adherence and treatment success in pulmonary tuberculosis patients.
Or
A health belief model-based motivational interviewing for medication adherence and treatment success in pulmonary tuberculosis patients.
- Page 1, line 25: Authors have used the “HBM-based MI intervention” more frequently in the text, therefore, it will be helpful to the readers if the same is used instead of “MI based on online HBM”.
- Page 3, Figure 1: Mention the inclusion criteria based on which 214 TB patients were selected for the study, either in the material and method section or as a supplementary.
- Page 5, Table 1: The layout of the table needs to improve for clarity.
Suggestion is to incorporate a column in front: 1. Gender, 2. Marital Status, 3. Education level, 4. Have Occupation, 5. Age, 6. Income
The authors need to mention in the text what is depicted in the bracket. In addition, it will be appropriate to provide the scoring scheme as a supplementary material.
- Page 6, Table 2: The layout of the table needs to improve for clarity.
Suggestion: Incorporate a column in front: 1. Adherence, 2. Treatment Success and mention what is represented in the brackets and present the scoring scheme as a supplementary material.
- Page 6, Table 3: The spelling of variable needs to be corrected. Secondly to maintain the consistency in the text, the authors can use “HBM-based MI intervention” instead of “MI-HBM intervention”.
- Page 6, Table 4: The layout of the table needs to improve for clarity.
Similar suggestion: Incorporate a column in front: 1. Age, 2. Education Level, 3. Income, 4. Knowledge of TB, 5. Family Support, 6. Perceptions of attitudes of health workers. Mention what is represented in the brackets and present the scoring scheme as a supplementary material.
- Page 6, Line 209, 211, 218: To aid in the understanding of variety of readers, it will be appropriate if authors mention ARR, CI, and RR. For instance, Absolute Risk Reduction (ARR), confidence interval (CI), and the relative risk (RR).
- Page 9 and 10: Some references need to be corrected. For instance, the spelling of tuberculosis in reference 1, 6, 23, 30, and 32 is mistakenly written as tuberkulosis.
Author Response
Dear reviewers 3,
For feedback points, please see the attachment.
Kind Regards
Ni Made Parwati
